# A Multi-Task Deep Learning Framework for Skin Lesion Classification, ABCDE Feature Quantification, and Evolution Simulation

## Abstract

Early detection of melanoma significantly improves survival rates, but many deep learning approaches do not justify their predictions with established dermatological assessment metrics. This work introduces a multi-task neural network that classifies skin lesions and quantifies interpretable ABCDE (Asymmetry, Border irregularity, Color variation, Diameter, Evolving) features. Trained on the HAM10000 dataset, the model achieves 89% accuracy overall and an AUC of 0.96 for the detection of melanoma in addition to providing quantitative scores for each characteristic. In addition, a module for lesion evolution visualizes a simulated ABCD feature trajectory and gives a more interpretable progression pattern from benign to malignant. Because HAM10000 contains only static images, the "E" (Evolving) feature was simulated computationally as it modeled the temporal trajectories of ABCD features in latent space. This improves diagnostic transparency and can assist dermatologists and educators by linking deep learning outputs to established clinical assessment criteria.

## 1 Introduction

Melanoma, an aggressive form of skin cancer, is one of the leading causes of death due to skin cancer [3]. Early diagnosis is important because the 5-year survival rate exceeds 90% for early-stage melanoma, but drops below 20% for advanced stages [3]. In order to differentiate between harmful and harmless lesions, dermatologists utilize the ABCDE method. "A" stands for "asymmetry," as malignant skin lesions often appear to be uneven; "B" stands for "border irregularity," as scientists search for jagged or notched edges; "C" stands for "color variation"; "D" stands for diameter, as larger lesions are more likely to be malignant; and "E" stands for "evolving," as skin lesions evolve over time [7]. If a lesion displays two or more of the attributes described above, the lesion is most likely harmful melanoma. The ABCDE criteria are effective because they are easy to understand and to screen for suspicious lesions [7].

Recent advances in medical imaging have made it possible to create realistic transformations of medical images. For example, Jütte et al. (2024) utilized a CycleGAN to create a sequence of dermoscopic images that show the potential of a benign nevus transforming into a malignant melanoma [3]. As discussed above, the quantification of ABCDE features changing over time as well as the actual images changing can improve our understanding about melanoma growth patterns [3].

In this work, a deep-learning framework that combines classification, ABCDE feature quantification, and feature evolution simulation is proposed.

Submitted to 39th Conference on Neural Information Processing Systems (NeurIPS 2025). Do not distribute.

## 2 Methodology

### 2.1 Overview of the Framework

The framework contains two main components: a CNN to perform lesion classification and ABCDE feature regression from a dermoscopic image, and also an evolution simulation module that shows how ABCDE features might progress over time. Given a dermoscopic image of a skin lesion, it is first optionally preprocessed (including lesion segmentation and color normalization). The multi-task CNN then processes the image to output both a class prediction and a set of numeric scores corresponding to A, B, C, and D features. The CNN is optimized by using a combined loss that includes classification error and regression error on the ABCDE scores. After this model is trained, it can provide an interpretation for its diagnosis by showing the ABCDE scores. For the evolution simulation, a lesion image is taken and a sequence of future images that shows increasing malignancy is generated. This CNN model is applied to each generated frame to track how the scores change.

### 2.2 Multi-Task CNN Architecture

This multi-task deep learning model is built based on a convolutional neural network that first extracts a shared representation of the input image, followed by two "heads" (output branches). These branches consist of one for lesion classification and one for ABCDE feature regression. ResNet50 was chosen as the backbone architecture due to its balance of depth and efficiency [2] [5]. The classification head is a dense layer that produces a probability distribution over the lesion classes. The regression head is a fully-connected dense layer to produce 5 values corresponding to [A, B, C, D, E]. Overall, the study uses linear outputs with appropriate activation/normalization; this is done to make sure that the feature values fall in a reasonable range. E is hardcoded as 0.0 because HAM10000 only contains static images, where only A, B, C, and D features can be analyzed. The images are rescaled to 224 by 224 pixels, and all the color channels are also normalized. The ResNet50 backbone processes the image through a series of convolutional layers. This gives a final feature map which is global-average-pooled to a 2048- dimensional feature vector. This vector represents high-level information about the lesion. Also, the network is trained to predict the ABCDE features, so the vector encodes information relevant to asymmetry, border, color, and others, in addition to other features useful to classify lesions. The classification head takes in the 2048 feature vector and produces logits for each of the seven classes for HAM10000 [6]. These include nv, mel, bcc, akiec, bkl, df, and vasc and correspond to melanocytic nevus, melanoma, basal cell carcinoma, actinic keratosis, benign keratosis, dermatofibroma, and vascular lesion [4]. A cross-entropy loss is used for this head during training. The regression head maps the same feature vector to five numeric outputs representing [A, B, C, D, E]. No activation (linear output) is applied for regression. However, these values are constrained through the training data scaling and loss function; this is so that the outputs remain in plausible ranges.

### 2.3 ABCDE Feature Engineering

- Asymmetry (A): The lesion's shape and color distribution are compared across the axes for asymmetry [3].

- Border Irregularity (B): An irregular border is one that is ragged, notched, or blurred. Two aspects are captured: the shape irregularity and the sharpness of the border [3]. For shape irregularity, the lesion's convex hull is computed and compared to the actual border [3].

- Color Variation (C): The amount of different colors and shades are measured in the lesion. Common criteria for skin lesion images include colors like light brown, dark brown, black, blue-gray, white, and red [3]. A melanoma often has different colors. To quantify this value, the dispersion of colors in the lesion is computed.

- Diameter (D): For the most part, lesions with a diameter greater than 6 millimeters are deemed as suspicious. However, these images lack a consistent physical scale as the zoom level varies [1]. HAM10000 images come from different devices and magnifications [4].

- Evolving (E): Because the dataset does not contain time-series images, the lesion evolution could not be directly measured or predicted. That is why this study focuses only on the static ABCD features for regression.

## 2.4    Model Training Strategy

The model is trained on the HAM10000 dataset [6]. During each training epoch, images are sampled such that each class is roughly equally represented. The data is split in this manner: 70% of the images for training, 10% for validation, and 20% for testing. For preprocessing, each image is resized to 224x224, the hair removal filter is applied, the color is normalized, and the lesions are segmented. All of these experiments use PyTorch and are trained on an NVIDIA A100 GPU.

## 2.5    Lesion Evolution Simulation

Here, how ABCD features might change over time is observed. The "time steps" would be simulated by a small network that adjusts this state in the direction of malignancy. At each time step, it updates the latent feature vector by predicting how it will change.

# 3    Results and Discussion

On the HAM10000 dataset, the multitask CNN model did end up showing a strong classification performance. The overall accuracy was 89% as it correctly classified 89% of all test samples. In the simulated ABCD score trajectory below, (Figure 1), A, C, and D all increase smoothly across the steps. As a lesion becomes more malignant, it typically becomes more asymmetric, shows greater variation in color, and increases in size. The upward trends in these scores suggest that the model captures these expected patterns of malignant progression. In contrast, the B score of border irregularity was completely flat near zero. The model struggles to predict this feature accurately most likely due to noisy or insufficient training labels for border irregularity. The lack of progression in B shows a current limitation in modeling that particular clinical feature.

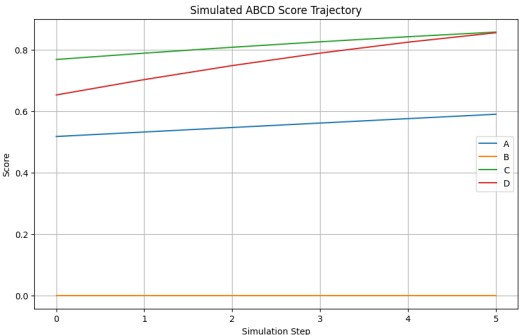

Figure 1: Simulated ABCD Score Trajectory

The results show that the multi-task CNN performs well overall as it combines strong lesion classification with interpretable ABCD feature predictions. It performs fairly well on key clinical tasks. For instance, with melanoma detection, it has an AUC of 0.96. It also learns to predict asymmetry, color variation, and diameter with strong accuracy. The model demonstrates that these features are not only predictable but also embedded meaningfully within the network's latent space. This is shown by the smooth trajectory and increasing malignancy indicators in the evolution simulation.

One clear limitation is the poor performance in predicting border irregularity (B). This likely stems from how B was labeled. It was most likely based on simple segmentation heuristics rather than clinical assessment. This introduced noise and weakened both regression and simulated trends. Also, the dataset's class imbalance affected both classification and regression accuracy for rare lesion types. Another limitation is that the evolution simulation was performed in latent feature space and not directly on images. This visual progression remains abstract.

This system could assist clinicians not only in diagnosing skin lesions but also in interpreting why the model made a decision. This is through the ABCD feature outputs. The evolution simulation can offer "what-if" previews of how lesions might progress toward malignancy, and this can support patient education and monitoring.

## 4    Negative Impact Statement

While this work aims to assist dermatologists in early melanoma detection, it carries potential risks if misused or misinterpreted. First, models trained on datasets such as HAM10000 may underperform on underrepresented skin tones, and this could reinforce healthcare inequities. Second, reliance on automated predictions without clinical oversight could lead to misdiagnoses or delayed treatment. Third, lesion evolution simulations may be misinterpreted as clinically verified progressions. Future deployments should ensure diverse data representation and make sure there as a human–AI collaboration rather than just automation.

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
