# OpenReview forum: "A Multi-Task Deep Learning Framework for Skin Lesion Classification, ABCDE Feature Quantification, and Evolution Simulation"
_EurIPS.cc/2025/Workshop/MedEurIPS — EurIPS 2025 Workshop MedEurIPS Submission_

### Official Review · Reviewer_aWCL · 2025-10-24
**Review for A Multi-Task Deep Learning Framework for Skin Lesion Classification, ABCDE Feature Quantification, and Evolution Simulation**

**Rating:** 6
**Confidence:** 3

**Review:**

Summary:
This paper proposes a multi-task deep learning framework combining skin lesion classification, ABCDE feature quantification, and evolution simulation. While the idea is promising for improving interpretability in medical imaging, the work still needs more convincing experiments and grounding to be truly compelling.

Strengths:
- Important motivation in solving the gap between black-box AI models and clinically interpretable diagnosis.
- Well-designed multi-task architecture that integrates classification and ABCDE feature regression for better explainability.

Suggesions:
- Include ablation or baseline comparisons to demonstrate the benefits of multi-task learning.
- Clarify reference sources and experimental details to enhance credibility.

---

### Official Review · Reviewer_UZKH · 2025-10-30
**A interesting concept for interpretable skin lesion classification with established dermatological criteria, but further validation and details are required.**

**Rating:** 4
**Confidence:** 4

**Review:**

This work introduces a multi-task neural network that simultaneously classifies skin lesions and quantifies ABCDE features from dermoscopic images.
### Pros
1.  The model provides skin lesion classification results with ABCDE scores, aligning its output with established dermatological criteria.
2.  The model achieves high classification performance with 89% overall accuracy and an AUC of 0.96 for melanoma detection, alongside ABCD scores.

### Cons
1.  The paper's claim to quantify the "Evolving" (E) feature is misleading. This feature is hardcoded to zero during training and is not a validated prediction based on temporal data. The authors could follow the work of Jütte et al. (2024) to generate temporal data for validation.
2.  The method is designed for a seven-class classification problem, but the paper only reports overall accuracy and melanoma AUC, which is insufficient. The evaluation lacks essential per-class metrics (e.g., precision, recall, F1-score) , making it impossible to assess performance on rare but critical lesion types.
3.  The paper lacks key design details for the Lesion Evolution Simulation. The description of a "small network" is vague, which makes the "Simulated ABCD Score Trajectory" less convincing.
4.  Critical training details are missing. It is unclear how the loss function is formulated and how the regression and classification losses are combined and weighted. For a multi-task model, this is a crucial detail for understanding performance and for reproducibility.

---

### Decision · Program_Chairs · 2025-10-31

**Decision:**

Reject

**Comment:**

Both reviewers find the idea of combining lesion classification with interpretable ABCDE feature quantification relevant and promising. However, they note missing implementation details, limited evaluation, and unclear handling of the “Evolving” feature. While conceptually interesting, the work requires stronger validation and methodological clarity.